# Spectral Imaging of UV-Blocking Carbon Dot-Based Coatings for Food Packaging Applications

Benedetto Ardini [1], Cristian Manzoni [2], Benedetta Squeo [3], Francesca Villafiorita-Monteleone [3], Paolo Grassi [2], Mariacecilia Pasini [3,*], Monica Bollani [2,*] and Tersilla Virgili [2]

[1] Dipartimento di Fisica, Politecnico di Milano, 20133 Milano, Italy; benedetto.ardini@polimi.it
[2] Istituto di Fotonica e Nanotecnologie-CNR, 20133 Milano, Italy; cristianangelo.manzoni@cnr.it (C.M.); grassi.paolo.95@gmail.com (P.G.)
[3] Istituto di Scienze e Tecnologie Chimiche "Giulio Natta" (SCITEC), Consiglio Nazionale delle Ricerche (CNR), 20133 Milano, Italy; benedetta.squeo@scitec.cnr.it (B.S.); francesca.villafiorita@scitec.cnr.it (F.V.-M.)
[*] Correspondence: pasini@scitec.cnr.it or mariacecilia.pasini@scitec.cnr.it (M.P.); monica.bollani@ifn.cnr.it (M.B.)

**Abstract:** Nowadays, there is an increased demand to develop alternative non-plastic packaging to be used in the food industry. The most popular biodegradable films are cellulose and poly(lactic acid) (PLA); however, there is still the need to increase their UV absorption to protect the packaging content. In this work, we have covered those biodegradable films with thin coatings based on carbon dots (CDs) dispersed in polyvinyl alcohol (PVA) deposited by spin- or spray-coating techniques. We report a strong increase in the UV light-absorbing properties, together with a detailed morphological characterization; moreover, we show the results of a new microscopy and spectral imaging technique applied to the coated samples. The scientific and technological novelty of this approach is the possibility of characterizing large areas of the material surface by the simultaneous detection of PL spectra in all the pixels of a highly spatially-resolved two-dimensional (2D) map of the surface. We report UV-excited PL maps whose detailed information allows us to clearly identify regions with different spectral behaviors and to compare their characteristic signals for different CDs:PVA deposition techniques.

**Keywords:** food packaging; carbon dots; hyperspectral imaging; cellulose; PLA

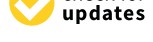



## 1. Introduction

Due to high production volumes, short usage time, and problems related to end-of-life management and release into the environment, single-use plastic packaging is an open question for the world [1]. As a result, there is an increased demand to develop alternative non-plastic packaging. Among the different types of packaging used in the food industry, biodegradable films such as cellulose and poly(lactic acid) (PLA) are selected in our work as primary packaging substrates to test the efficacy of an organic coating to improve their chemical-physical properties [2,3]. PLA and cellulose, indeed, are two of the market's most promising biocompatible food packaging materials due to their composability and overall properties such as biocompatibility and transparency and mechanical properties that are comparable with those of conventional polymers similar to those of oil-based counterparts [4–9]. However, these biodegradable films suffer from some drawbacks that make them unsuitable, for example, for long shelf-life foods. Among others, surface properties (i.e., poor adhesion properties) and poor barrier performances against gases, vapors, and UV radiation are the most relevant limitations that deserve improvement [10–12]. Carbon dots (CDs), part of the nanocarbon family, are based on a combination of aromatic and aliphatic regions with sizes around 10 nm. The constituents are graphene, graphene oxide, and diamond, obtained after reactions such as condensation, polymerization, carbonization, passivation, and crosslinking [13–18]. Recently, CDs have attracted considerable attention

as biocompatible and sustainable fluorophores with various applications [19–21]. CDs have very attractive photo-induced properties, such as broad absorbance, large photo-luminescence (PL), strong photothermal coefficients, and photoelectron transfer [22–25]. Furthermore, their chemical inertness, photostability, and biocompatibility make them suitable for novel multifunctional fluorescent probes in various biomedical applications, including optical imaging, drug delivery monitoring, biosensing, and phototherapy [26–29]. Moreover, CDs have been successfully applied in active and smart food packaging [30–33]. In this work, we propose, by exploiting the optical characteristics, chemical resistance, and low toxicity of CDs, a method to improve the UV radiation barrier of commercial packaging: it is based on the deposition of CD-based thin organic layers by spin- or spray-coating techniques. A morphological and optical characterization of the resulting coatings is reported, while, for the first time, a new microscopy and spectral imaging technique is proposed for their characterization. The scientific and technological novelty of this approach arises from the possibility of characterizing large material areas by the detection of PL spectra of each element (or pixels) of a highly spatially-resolved two-dimensional (2D) map of the sample surface. Moreover, we report UV-excited PL maps whose spectral details allow us to clearly identify areas with different behaviors, whose spatial and spectral properties depend on the deposition techniques.

Through these studies, we demonstrate that our deposition and characterization techniques contribute to making a step forward in many aspects: (i) by complying with the new circular economy principles aiming to overcome the environmental impact of conventional plastics from non-renewable resources, (ii) by going beyond the state of the art of packaging by engineering PLA and cellulose films with CDs:PVA to improve the UV radiation barrier, (iii) by applying novel spectral microscopy tools to assess both morphological features and light–matter interaction mechanisms on the coated surface of the packaging.

## 2. Spectral Microscopy Technique Applied to Coated Flexible Films

Spectroscopy is a technique to retrieve the physical properties of matter by studying the electromagnetic spectrum of light reflected, scattered, transmitted, or emitted by objects. The spectrum, which is the intensity as a function of the light wavelength (or frequency), is collected by means of a spectrometer typically based on a dispersive element (e.g., grating, prism) that spatially separates the spectral components, subsequently detected by an array of photodetectors [34]. A dispersive spectrometer typically filters the light by means of an input slit, which inevitably reduces the throughput of the device. Another class of spectrometers relies on the Fourier Transform (FT) approach. In FT spectroscopy, an interferometer splits the optical waveform into two delayed replicas, whose integrated energy is subsequently measured by a point detector as a function of their delay. This produces the interferogram, whose FT yields the intensity spectrum of the waveform according to the Wiener–Khinchine theorem [35]. The advantages of FT spectroscopy with respect to dispersive methods are as follows: (i) higher light throughput determined by the absence of slits (Jacquinot's advantage), (ii) flexible spectral resolution inversely proportional to the maximum scanned delay, (iii) possibility to record multiple spectra from a two-dimensional field of view (FOV) in parallel.

The challenge of FT spectroscopy is the need to stabilize the delay introduced by the interferometer with a precision better than 1/100 times the optical cycle, which, for visible light @600 nm wavelength, corresponds to 20 attoseconds: this is the reason why this technique is mainly exploited to operate in the infrared region where the optical cycle is much longer, thus relaxing the stability requirements.

Spectral imaging (SI) combines spectroscopy with an imaging system to measure the spectrum for each element (or pixel) of the image. This produces three-dimensional datasets, called spectral cubes, characterized by one spectral and two spatial coordinates. Such datasets can be represented as two-dimensional maps in which each pixel contains a spectrum or, equivalently, as stacks of images with each one at a selected wavelength.

Different SI approaches have been developed starting from the second half of the last century (see a list of SI techniques in Supplementary Materials). When SI is performed by FT spectroscopy, the FT operation produces a continuous spectrum, leading to the so-called hyperspectral imaging (HSI) method: the technique is also referred to as Fourier Transform HSI (FT-HSI). In contrast to dispersive HSI methods, FT-HSI offers the unique advantage of acquiring the spectra of all image pixels in parallel, enabling widefield operation.

Recently Candeo et al. [36] developed a compact FT-based microscope (FT-HSM) based on TWINS (Translating-Wedge-Based Identical Pulses eNconding System), an ultrastable common-path interferometer [37]. The high interferometric precision and exceptionally long-term stability of the TWINS combined with the advantages of FT-HSI make the FT-HSM a suitable system to detect widefield PL maps with high spatial and spectral resolution and high SNR.

## 3. Materials and Methods

### 3.1. Samples Preparation

*CDs:PVA synthesis.* The material is prepared according to the literature [38]. In more detail, polyvinyl alcohol (PVA) (1 g, Sigma-Aldrich, St. Louis, MO, USA, average mol wt 30,000–70,000), citric acid (0.1 g, Tokyo Chemical Industry, Tokyo, Japan), and branched polyethylene imine (0.1 g, Sigma-Aldrich, Merk Life Science S.r.l., Milano, Italy, average mw 800) were mixed with 9 mL of milli-Q water and stirred until complete dissolution. The precursor solution was transferred in a Pyrex tube, and the solvothermal synthesis was conducted in a microwave system (Discover 2.0, CEM S.r.l, Cologno Al Serio, Bergamo, Italy). The reaction was run for 2 h at 135 °C. The as-obtained solution was used without further purification as reported by Hess, Samuel C. et al. [38].

*Film preparation.* Two different substrate films based on cellulose and poly(lactic acid) (PLA) are provided by Corapack (Corapack Srl, Brenna, Como, Italy) in the framework of the Regione Lombardia Spatial project. The thickness of C and PLA films are 23 μm and 30 μm, respectively [10]. PVA:CDs suspensions were deposited on $2 \times 2$ cm$^2$ flexible films of PLA or cellulose and fixed on glass slides with adhesive tape by spin- or spray-coating. Specifically, the spin-coated samples were prepared by depositing twice 200 μL of the as-prepared PVA:CDs suspension on each sample at 1000 rpm for 120 s. The spray-coated samples were instead prepared after diluting with water the starting suspension to 50% *v/v*. The new suspension was then transferred to a spray jar and sprayed horizontally on each substrate. After the spraying, the samples were left at room temperature until complete water evaporation.

Figure 1 depicts the preparation process.

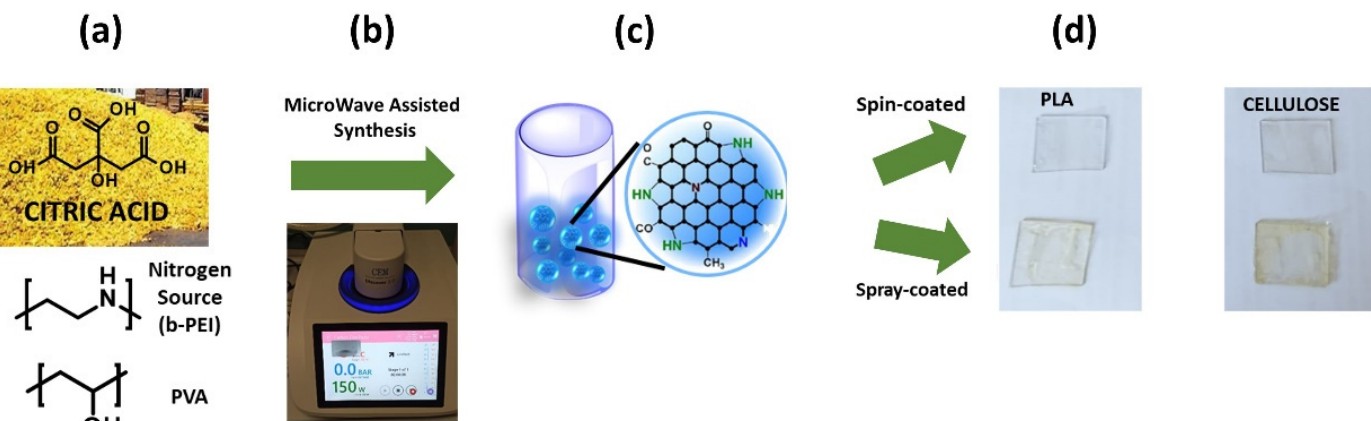

**Figure 1.** Schematic representation of the CDs:PVA-coated films preparation: (**a**) Citric acid, b-PEI, and PVA are mixed; (**b**) synthesis assisted by MicroWave; (**c**) formation of CDs; (**d**) PLA and cellulose films coated with the spin- or spray-coating techniques.

### 3.2. Sample Characterization

*UV-Vis Spectroscopy.* The optical properties of the films were recorded at room temperature using a Perkin Elmer UV/VIS/NIR Lambda 900 (Milano, Italy) spectrometer. The measurement provided the transmission averaged over an area of approximately 1 cm$^2$.

*AFM and SEM.* First, we systematically studied the morphology of the coated flexible food packaging films by a combination of atomic force microscopy (AFM) (Innova AFM of Veeco company, Binasco, Italy) and scanning electron microscopy (SEM) (Philips XL30 S-FEG SEM, Milan, Italy). While by SEM, indeed, the uniformity of the coating deposition is observed and the roughness can be only qualitatively characterized [39], a nanoscale analysis of the morphological roughness of the film is provided by AFM [10]. AFM topography images are acquired in tapping mode, using a super-sharp silicon probe (typical radius of curvature 2 nm). The root mean square (rms) roughness on a flat area is calculated as the standard deviation of the topography by Gwyddion software (version 2.55, 2019, Czech Metrology Institute, Brno, Czech Republic) described in [40].

*FT Hyperspectral Microscopy.* We have studied the UV-excited PL emitted by the CDs-based coatings by performing measurements with the FT-HSM in the epi-illumination scheme depicted in Figure 2. The excitation light provided by a mercury lamp (Leica HG 50 W/AC Type 307-072.058, Wetzlar, Germany) is filtered by a bandpass filter (BPF, Chroma ET365/10x, Bellows Falls, VT, USA) selecting a 10 nm band around 365 nm wavelength. It is then reflected by a dichroic mirror (DM, Chroma ZT 387, Bellows Falls, VT, USA) and focused on the sample by a 20× microscope objective (Leica PL FLUOTAR BD, 20×, N.A. 0.4, Wetzlar, Germany). The PL from the sample is collected by the same objective and transmitted by the DM on the detection branch where a low-pass filter (LPF, Thorlabs FGL400, Newton, NJ, USA) removes the residual excitation light. The PL map is finally imaged on a 2D detector (Luca R, Andor, Belfast, Northern Ireland, size 8 × 8 mm$^2$, 1002 × 1004 pixels, 14-bit depth, spectral sensitivity from 400 nm to 1100 nm) by means of the tube lens of the microscope. The TWINS interferometer (developed at Politecnico di Milano and Istituto di Fotonica e Nanotecnologie-CNR, Milano, Italy) is located between the tube lens and the 2D detector. It is based on two wedges of YVO$_4$ birefringent crystal (apex angle $\alpha = 10°$, transverse size 30 mm), one of which is laterally translated by a linear motorized stage (Physik Instrumente M-112.12S1, Karlsruhe, Germany, minimum step size: 0.02 μm).

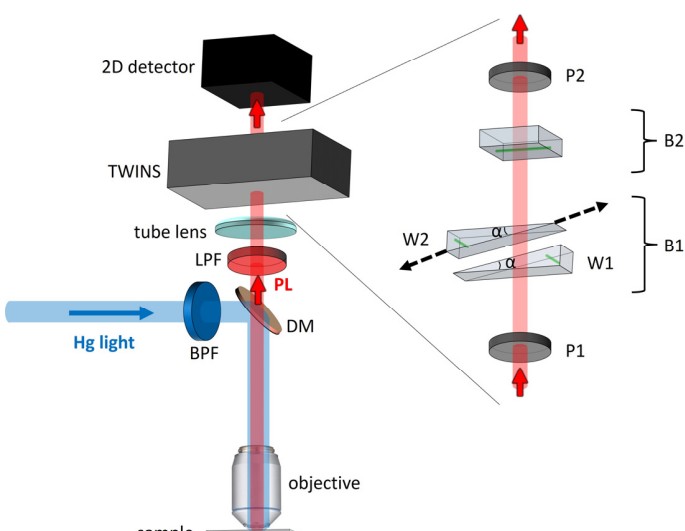

**Figure 2.** Scheme of the FT-HSM. Hg light: mercury lamp excitation light; PL: sample photoluminescence; LPF: low-pass filter; BPF: bandpass filter; DM: dichroic mirror; P1: first polarizer; B1: first YVO$_4$ block shaped into two wedges with apex angle $\alpha$; W1: fixed wedge; W2: translating wedge; B2: second YVO$_4$ block; P2: second polarizer. The black dashed arrows indicate the wedge W2 translation direction. The green lines indicate the YVO$_4$ crystal axis in the two wedges W1 and W2 of block B1 and in block B2.

For each measurement, we acquired a set of images at 100 phase delays, imposed by the TWINS. The wedge was translated ±200 μm with respect to the zero phase-delay condition: this corresponds to a delay range of ±65 fs, leading to the spectral resolution of 22 nm @600 nm after FT. The sampling step is 4 μm, corresponding to 0.65 fs, smaller than the maximum step imposed by the Nyquist–Shannon limit (0.67 fs) for light at 400 nm.

## 4. Results and Discussion

### 4.1. Morphological Studies by AFM and SEM

Figure 3 reports the results obtained by AFM and SEM measurements. Tapping-amplitude AFM images of two cellulose films treated with the CDs solution deposited by spin- and spray-coating techniques (respectively shown in Figure 3a,b) clearly highlight depressions in the samples. As these depressions have not been observed in absence of organic coatings [10], we infer that their presence in the coated films is probably due to the entrapment of air and its slow release due to the viscosity of the solution used for film preparation [41]. In both spin- and spray-coating, the rms roughness calculated on the flat areas is less than 4 nm ± 0.5 nm, while the depressions, presenting a circular geometry with diameters spanning from ~800 nm to ~3 μm, are between 70 nm and 90 nm deep.

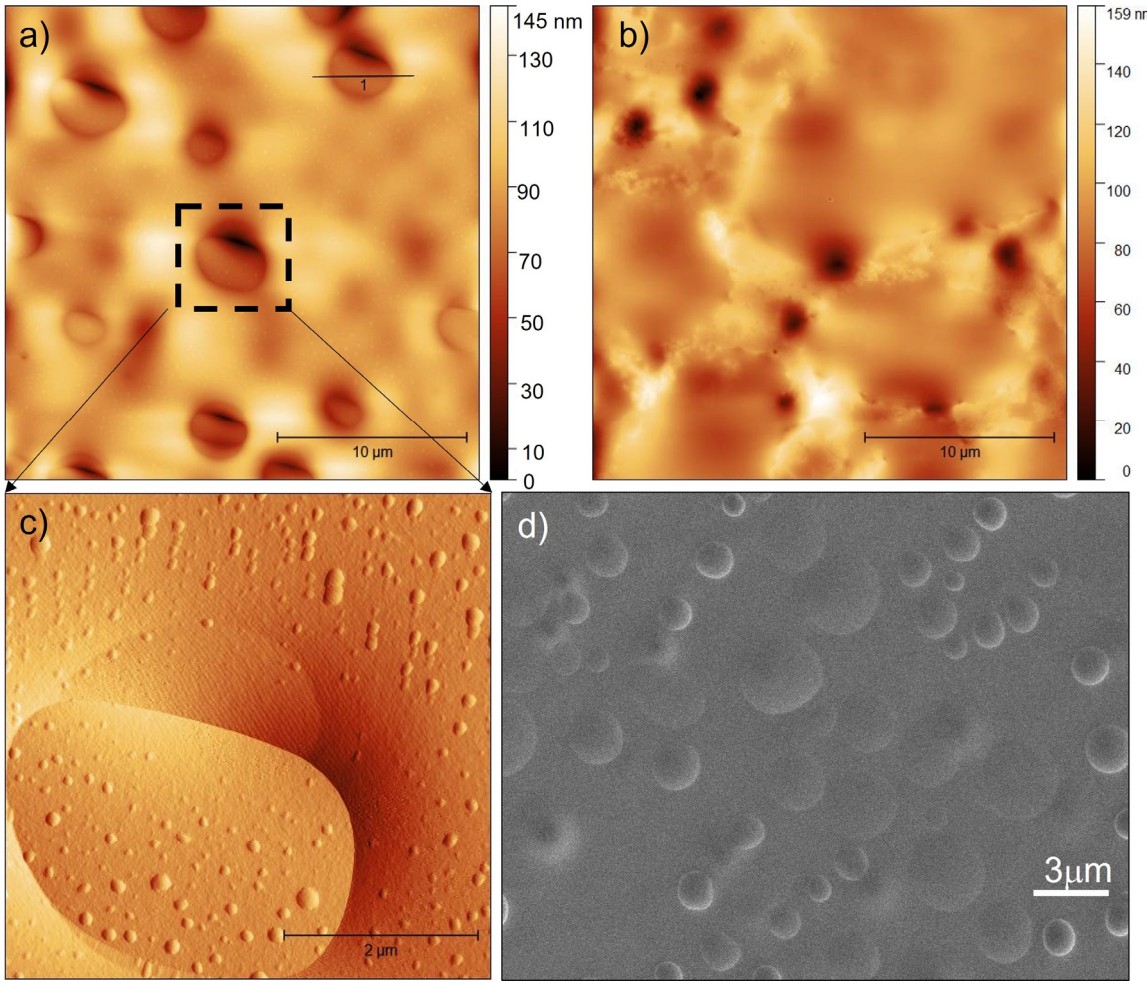

**Figure 3.** (**a**) Topography AFM image achieved in tapping mode of a spin-coated cellulose film (the dashed-line rectangle identifies the area shown in panel (**c**)). (**b**) Topography AFM image achieved in tapping mode of a spray-coated cellulose film. (**c**) High-resolution tapping amplitude AFM image obtained by scanning the area indicated in panel (**a**). (**d**) SEM image of a spin-coated cellulose film.

By focusing on a small area, it is possible to identify smaller depressions with a diameter of ~100 nm (Figure 3c).

Several samples of cellulose and PLA treated with identical solutions of PVA:CDs confirm the formation of similar corrugations. Here we report a SEM image (Figure 3d) of a spin-coated cellulose sample that confirms the presence of the depressions highlighted in the AFM measurements.

### 4.2. Transmission and Photoluminescence Results

By measuring the transmittance spectra of films before and after the coating process (see Figure 4), we verified that, despite the presence of the above-mentioned depressions, the overall protective properties of these films in the UV range from 300 nm to 400 nm are effective.

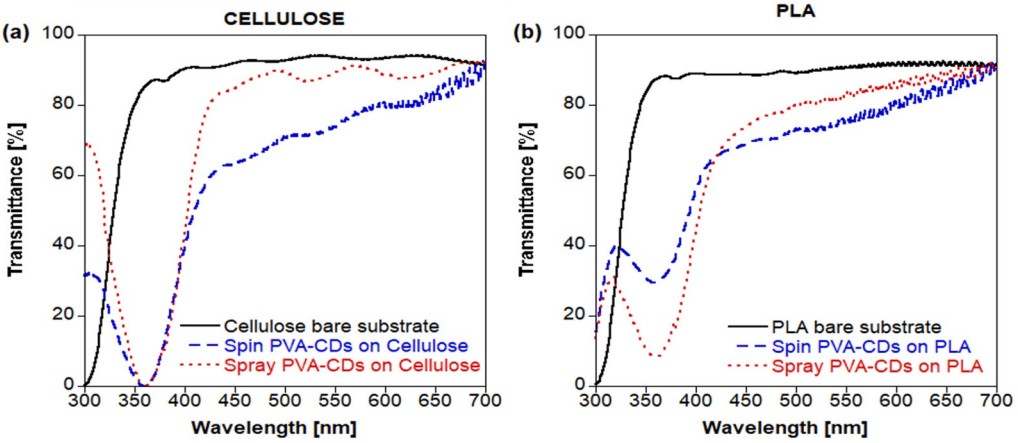

**Figure 4.** Transmittance spectra of (**a**) cellulose and (**b**) PLA bare substrates (black line), spin-coated films (dashed blue line), and spray-coated films (dotted red line).

The black solid lines in Figure 4 represent the transmission spectra of the cellulose and PLA bare substrates. The bare substrates transmit at least 90% of the light in the spectrum between 350 nm and 700 nm, while the light transmission in a broad band around 360 nm drops to 0% on both the spin-coated (blued dashed line) and the spray-coated (red dotted line) cellulose substrates (Figure 4a) and to 10% and 30% on the spray-coated (red dotted line) and spin-coated (blued dashed line) PLA samples, respectively (Figure 4b). Hence, the comparison between the transmittance spectra demonstrates that the coated cellulose films ensure better UV screening with respect to the coated PLA ones.

To assess a space-resolved widefield characterization of the coated samples, we acquired PL images with the FT-HSM, following the procedure described in Section 3.2. Table 1 reports the integration time for the acquisition of each phase-delay image, for all the examined samples.

**Table 1.** Integration time for one frame set for each sample.

| Substrate | Coating | Integration Time (s) |
|---|---|---|
| Cellulose | spin | 3 |
| Cellulose | spray | 1.2 |
| PLA | spin | 0.5 |
| PLA | spray | 1.6 |

The results of the HSM measurements are shown in Figure 5 for the cellulose-based samples, and in Figure 6 for the PLA-based ones. In all cases, we are able to identify two regions, characterized by different PL spectra, respectively shown in panels (a) and (d). Region B is characterized by elements with circular shapes, while Region A identifies the

remaining area of the film surface. As such morphology is coherent with what has been observed in the AFM and SEM measurements discussed before (Figure 3a,d), we assert that the circular shapes are depressions in the film. The same morphological features were confirmed by preliminary measurements with a confocal microscope (see Supplementary Materials). The PL spectra present a peak at around 500 nm. Within each of the four films, the PL peaks in regions A and B are shifted by a few tens of nanometers, denoting different material aggregations.

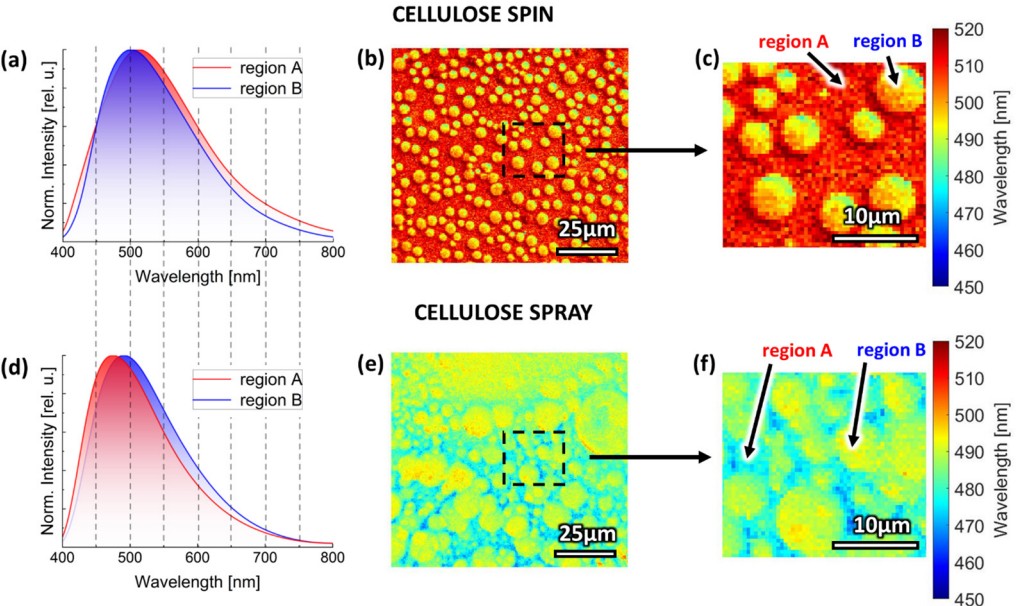

**Figure 5.** (**a**,**d**) Normalized PL spectra obtained by averaging an area of 50 × 50 μm² of spin- and spray-coated cellulose samples' surface. (**b**,**e**) Map of the PL peak wavelength in each point of the image. (**c**,**f**) Zoom of the dashed areas, respectively shown in panels (**b**,**e**). Panels (**b**,**c**,**e**,**f**) are plotted with the same color scale.

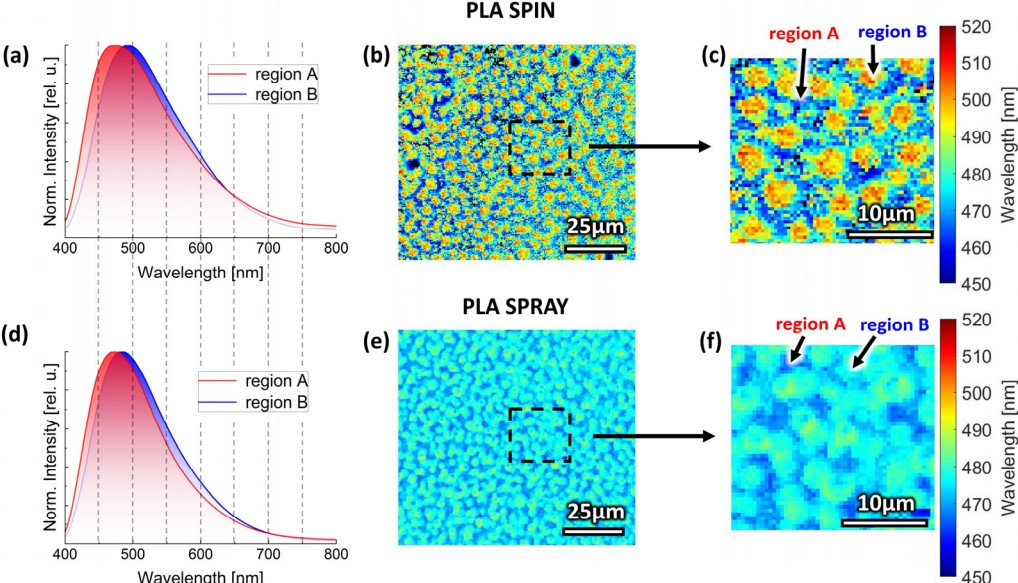

**Figure 6.** (**a**,**d**) Normalized PL spectra obtained by averaging an area of 50 × 50 μm² of spin- and spray-coated PLA samples' surface. (**b**,**e**) Map of the PL peak wavelength in each point of the image. (**c**,**f**) Zoom of the dashed areas, respectively shown in panels (**b**,**e**). Panels (**b**,**c**,**e**,**f**) are plotted with the same color scale.

Quantitative information can be drawn also by comparing the PL intensities from each sample (Figure 7). The overall PL intensity is higher in the spray-coated samples, independently from the substrate, which confirms [34] that a spray-coated film is typically thicker than a spin-coated one. Moreover, in all the films, the PL is stronger in region B (depressions on the surface) with respect to A. We attribute this effect to a PL amplification mechanism occurring in the depression areas. More measurements are on the way to better understand this effect, which is beyond the scope of the present paper.

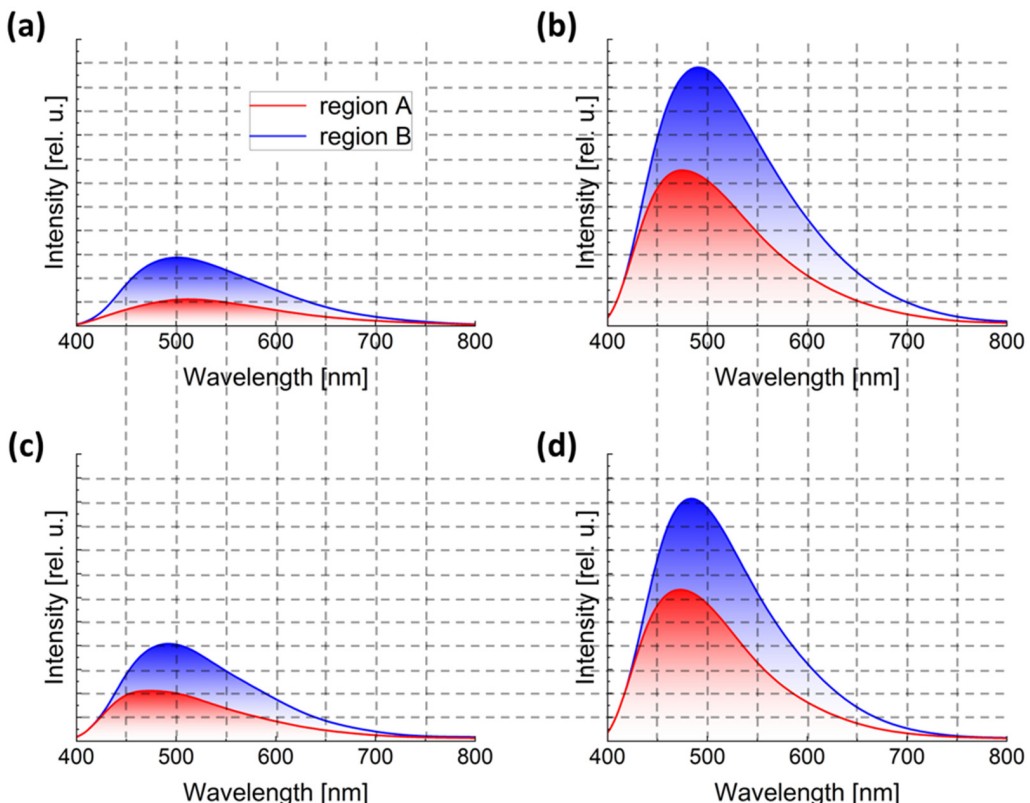

**Figure 7.** PL spectra in region A (red line) and B (blue line) obtained by averaging over an area of $50 \times 50 \ \mu m^2$ in all samples. Spectra are normalized on measurement integration time (see Table 1). (**a**) cellulose spin, (**b**) cellulose spray, (**c**) PLA spin, (**d**) PLA spray.

## 5. Conclusions

In this work, we studied the morphological and spectral properties of the most popular biodegradable packaging films, i.e., PLA and cellulose, spin- or spray-coated with a CDs-based solution. We performed a morphological characterization, followed by spectroscopic measurements. The morphological analysis has revealed the presence of circular depressions. Transmission measurements have confirmed that the coating strongly increases the UV light barrier, blocking most of the light around 350 nm: the best performances are obtained with the cellulose substrate. By means of an HSM technique, we collected PL 2D maps over large sample surfaces. By proper spectral analysis, we identified in each sample two different regions, one of them associated with the presence of depressions. We believe that these results will be pivotal to a novel approach to increase the UV barrier in food packing applications, highlighting the potential of the novel spectral imaging technique as a rapid and scalable tool for flexible film characterizations.

**Supplementary Materials:** The following Supplementary Materials can be downloaded at: https://www.mdpi.com/article/10.3390/coatings13040785/s1, Figure S1: Fluorescence microscopy images of PVA:CDs-based spin-coated films on cellulose. References [42–44] are cited in the supplementary materials.

**Author Contributions:** Conceptualization, T.V., M.P. and M.B.; Materials and film preparation P.G., M.P., B.S. and F.V.-M.; methodology, C.M. and B.A.; software, C.M. and B.A.; validation, T.V., M.P. and M.B.; investigation, B.A., P.G., B.S. and F.V.-M.; data curation, B.A.; writing—original draft preparation, B.A., C.M., T.V., M.P. and M.B.; writing—review and editing, B.A., C.M., T.V., M.P. and M.B.; supervision, C.M., T.V., M.P. and M.B. All authors have read and agreed to the published version of the manuscript.

**Funding:** This work was supported by the sPATIALS3 project financed by Regione Lombardia; Agritech project (Centro Nazionale per le Tecnologie dell'Agricoltura-PNRR 2022-25); EPOCALE project financed by CNR (2022-2024); Functional carbon dots for ENhancing tomato production In a Circular Economy scheme (FENICE), CARIPLO Foundation grant number 2021-0626.

**Institutional Review Board Statement:** Not applicable.

**Informed Consent Statement:** Not applicable.

**Data Availability Statement:** Data sharing is not applicable to this article.

**Conflicts of Interest:** The authors declare no conflict of interest.

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
