# Peer review of "Spectral Imaging of UV-Blocking Carbon Dot-Based Coatings for Food Packaging Applications"

_coatings, doi:10.3390/coatings13040785_

Round 1

Reviewer 1 Report

The MS is well organized except for some references over 5 years and the references are not enough (less than 20).

Author Response

We thank the reviewer of the positive assessment and the precise feedback. The article has been implemented with new and recent references.

Reviewer 2 Report

This manuscript (coating-2325343) reported on the morphological and spectral properties of PLA and cellulose films and characterized by UV-Vis, AFM, SEM, FT-HSM. However, there are still some issues to be addressed before its acceptance.

1.      More references should be added in the first paragraph of the introduction.

2.      In the background of the introduction some literature could be presented on the use of CDs by others in the field of food packaging.

3.      What is the amount of spray-coating? Is it the same as spin-coating?

4.      More introduction on the development of the materials in packaging should be further clarified with some recent supporting articles: Biobased materials for food packaging; Packaging and degradability properties of polyvinyl alcohol/gelatin nanocomposite films filled water hyacinth cellulose nanocrystals; Electrospun Functional Materials toward Food Packaging Applications: A Review; etc.

5.      For better comparison, SEM images of spin-coated PLA film should be provided.

6.      The section numbering should be carefully rechecked. There are some mistakes on the numbering.

7.      More details on the raw materials about PLA and cellulose film should be provided, such as the source and thickness.

8.      There are still some minor typos and grammar issues. Authors should carefully recheck the whole manuscript.

9.      Some references out of the recent five years should be removed or replaced by those recent three years to present the novelty and timely of this work.

Author Response

1. More references should be added in the first paragraph of the introduction.

Following referees’ suggestions, we added three references in the first paragraph.

  1. In the background of the introduction some literature could be presented on the use of CDs by others in the field of food packaging.

Following referees’suggestions, we added new references in the first paragraph.

  1. What is the amount of spray-coating? Is it the same as spin-coating?

In the manuscript (line 132) we report: “The spray-coated samples were instead prepared after diluting with water the starting suspension to 50% v/v. The new suspension was then transferred to a spray jar and sprayed horizontally on each substrate”. Therefore, the spray-coated samples have about half the concentration of PVA:Cdots than the spin-coated samples, but we used a volume of 200 µl of the diluted solution for a 2×2 cm2 area.

  1. More introduction on the development of the materials in packaging should be further clarified with some recent supporting articles: Biobased materials for food packaging; Packaging and degradability properties of polyvinyl alcohol/gelatin nanocomposite films filled water hyacinth cellulose nanocrystals; Electrospun Functional Materials toward Food Packaging Applications: A Review; etc.

We thank the reviewer, and some new references are included also in this section.

  1. For better comparison, SEM images of spin-coated PLA film should be provided.

Unfortunately, the PLA-based material charges easily and, although using some special holders, it is not possible to characterize this type of coated-substrate via SEM.

  1. The section numbering should be carefully rechecked. There are some mistakes on the numbering.

We thank the referee and the section numbering have been changed.

  1. More details on the raw materials about PLA and cellulose film should be provided, such as the source and thickness.

We have added this information at page 3 line 122.

  1. There are still some minor typos and grammar issues. Authors should carefully recheck the whole manuscript.

Thanks to the referee we have read carefully the manuscript to correct typos and grammar issues.

  1. Some references out of the recent five years should be removed or replaced by those recent three years to present the novelty and timely of this work.

We have replaced reference 3 with a more recent one:

  1. Rajeswari, E. Jackcina Stobel Christy, E. Swathi, Anitha Pius Fabrication of improved cellulose acetate-based biodegradable films for food packaging applications Environmental Chemistry and Ecotoxicology

Volume 2, 2020, Pages 107-114

and reference 5 with:

Pedroni, M.; Vassallo, E.; Aloisio, M.; Brasca, M.; Chen, H.; Firpo, G.; Ghezzi, F.; Morandi, S.; Pietralunga, S.M.; Silvetti, T.; et al. Plasma sputtered tungsten oxide thin film on poly(lactic acid) for food packaging applications. Coatings 2021, 11, 1281.

Reviewer 3 Report

The study is well organized and good presented. Some minor revisions must be done before the acceptance for publication. Here are some of my minor comments

1. Separate the experimental result is section and subsections according to the journals author's guide.

2. Add the instrumental information for AFM and SEM used.

Best regards!

Author Response

  1. Separate the experimental result is section and subsections according to the journals author's guide.

Following the referee suggestion, we have added the subsections 4.1 and 4.2

2. Add the instrumental information for AFM and SEM used.

The instrumental information is added.